# LEARNING TRANSITIONAL SKILLS WITH INTRINSIC MOTIVATION

## ABSTRACT

By maximizing an information theoretic objective, a few recent methods empower the agent to explore the environment and learn skills without extrinsic reward. However, when considering to use multiple consecutive skills to complete a specific task, the transition from one to another cannot guarantee the success of the process due to the evident gap between skills. In this paper, we propose to learn transitional skills (LTS) in addition to creating diverse primitive skills. By introducing an extra latent variable for transitional skills and a compensation term for intrinsic reward, our LTS method discovers both primitive and transitional skills by maximizing the mutual information with compensation. By considering various robotic tasks, our results demonstrate the effectiveness of LTS on learning both diverse primitive skills and transitional skills, and show its superiority in smooth transition of skills over the state-of-the-art baseline DIAYN. In addition, we further show the significance of transitional skills for learning downstream tasks.

## 1 INTRODUCTION

Deep reinforcement learning (DRL) has shown its great effectiveness in learning various reward-driven skills in wide domains, such as performing robotic manipulation tasks (Levine et al. (2016)), playing video games (Mnih et al. (2015)), playing adversarial board games (Silver et al. (2016)) and implementing robot navigation in complex environments (Wang et al. (2018)). Nevertheless, for the majority of real applications, there is no reward in a long term until the agent reaches a goal state (Wu & Chen (2007)), especially in unseen environments. In such cases, DRL has difficulty in carrying out the tasks.

By observing the human intelligence that can explore their surroundings and learn valuable skills without reward, a couple of prior works have been recently proposed to generate skills without supervision by incorporating intrinsic motivation into DRL methods (Barto (2013),Ryan & Deci (2000)). Diverse skills can be autonomously acquired without extrinsic reward by maximizing an information theoretic objective using a maximum entropy policy (DIAYN (Eysenbach et al. (2018)); VIC (Gregor et al. (2016)); DAS (Sharma et al. (2019))). Discovered skills may help the exploration in complex environments, and can also serve as primitive skills for hierarchical DRL. Particularly, a high-level meta-policy could be adopted in the hierarchical framework to choose corresponding low-level primitive skills to complete tasks in order.

Although discovered skills are both distinguishable and diverse, it is still exceedingly difficult to integrate such skills for a complex task that requires smooth transitions between skills (Lee et al. (2018)). Take the basketball as an example: learning the passing, catching and shooting skills in an isolated way cannot guarantee to score in the court due to the possible failure in the process of transitions between skills. To address this problem, we propose to further learn transitional skills (LTS), where discovered primitive skills, same as prior works (Eysenbach et al. (2018)), are distinguishable and as diverse as possible.

More concretely, our LTS method learns both primitive and transitional skills by optimizing an information theoretic objective, where extra transitional skills are generated to fill in the gap between diverse primitive skills. For such purpose, aside from using the latent variable on which we condition primitive skills, an extra latent variable is introduced on which we condition our transitional policy. Furthermore, a compensation term has to be considered in the objective function because the

distinct between primitive skills and transitional skills will lead to the decline of multual information between the latent variable corresponding to primitive skills and states generated by transitional skills. This compensation considers the divergence of the latent variable corresponding to primitive skills and that for transitional skills.

Different from learning primitive skills, our learning process considers two arbitrary primitive skills and multiple transitional skills between them, where both primitive and transitional skillls are unknown and to be learned. By maximizing the multual information with compensation, both primitive skills and transitional skills are discovered, which can be used to effectively learn downstream tasks. On four simulated robotic tasks, experimental results show that our LTS can discover both primitive skills and transitional skills, successfully perform the transition between primitive skills that are distinguishable, and achieve a better peformance in comparison to the state-of-the-art baseline DIAYN.

The main contributions of our work can be summarized as follows. Our proposed LTS can learn both primitive and transitional skills without extrinsic reward, where the primitive skills are distinguishable and diverse, and the transitional skills can accomplish smooth transitions between primitive skills. And extensive experiments are conducted, which demonstrates the effectiveness of our LTS method in solving downstream tasks, performing the transition between primitive skills as well as the weighted way to compose skills.

## 2 PRELIMINARIES

**RL**: In the standard RL setup, an agent interacts with an environment over discrete time. At time step $t$, the agent observes the current state $s_t$ and selects an action $a_t$ according to a policy $\pi(a_t|s_t)$. Then, the agent receives a reward $r_t$ and comes to the next state $s_{t+1}$. The objective of learning is to maximize the discounted return $R = \sum_{t=0}^{\infty} \gamma^t r_t$ of the policy $\pi$, where $\gamma \in [0, 1]$ is a discount factor.

**Learn Skills with RL**: Using the notation from information theory: we introduce two random variables $S$ and $A$ for states and actions, respectively. To discover diverse skills, a latent variable $\omega_n \sim p(\omega)$ is defined on which we condition the policy $\pi(a_t|s_t, \omega_n)$. Such policy is defined as a *skill* (Eysenbach et al. (2018)). Prior works verify that maximizing the mutual information between the states $S$ and the skills $\omega_n$ will successfully generate lots of distinguishable, diverse and useful skills.

By primarily maximizing the mutual information between the final state $S_f$ and the skills $\omega_n$ given the initial state $S_0$,

$$\mathrm{I}(S_f; \omega_n|S_0)^1,\tag{1}$$

the variational intrinsic control (VIC) (Gregor et al. (2016)) shows the success of acquiring distinguishable skills from the final state. Furthermore, in order to enhance the diversity of skills as much as possible, DIAYN primarily maximizes the mutual information between the state $S$ at all time stamps and the skills $\omega_n$ (Eysenbach et al. (2018)),

$$\mathrm{I}(S_t; \omega_n) + \mathrm{H}[A|S, \omega_n]^2,\tag{2}$$

which indicates that different skills generate different trajectories that traverse different states, and such diverse skills can be identified distinguishably.

Both VIC and DIAYN successfully discover primitive skills by maximizing the mutual information betweens state and skills. However, prior works cannot accomplish a variety of robot tasks that require smooth transition between skills. To address such problem, we propose the LTS scheme in this paper to learn both primitive and transitional skills by using an information theoretic objective function.

---

[1]The mutual information is denoted as the formation of conditional probability and contains the initial observation $s_0$: $\mathrm{I}(S_f; \omega|S_0) = -\sum_{s_f} p(s_f|s_0) \log p(s_f|s_0) + \sum_{\omega, s_f} p^J(s_f|s_0, \omega) p^C(\omega|s_0) \log p^J(s_f|s_0, \omega)$. The controllability distribution $p^C(\omega|s_0)$ maximizes the behavior diversity.

[2]The second term suggests that each skill should act as randomly as possible, aiming improving the exploration.

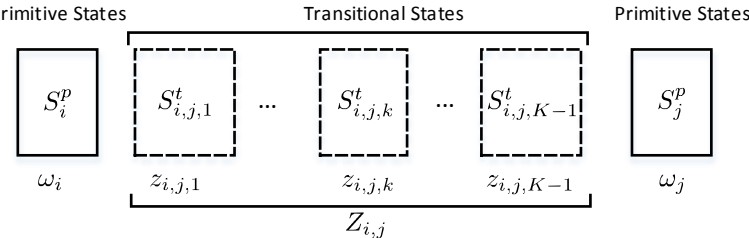

Figure 1: Block diagram of primitive and transitional skills.

## 3 METHODOLOGY

In this section, we elaborate our proposed LTS method to discover both primitive and transitional skills without extrinsic reward. Define by $S$ and $A$ states and actions, respectively. Define by $\Omega_n \sim p(\omega)$, $1 \leq n \leq N$, $N$ latent variables on which we condition the primitive skill $\pi(a_t|s_t, \omega)$. Given the starting and ending primitive skill that are conditioned on $\omega_i$ and $\omega_j$, respectively, we define $z_{i,j,k}$, $1 \leq k \leq K-1$, as $K$ latent variables on which we condition the transitional skill $\pi(a_t|s_t, z_{i,j,k})$.

Figure 1 shows the block diagram of primitive and transitional skills as well as the correponding states. Our objective is to discover primitive skills $\pi(a_t|s_t, \omega_i)$, $\pi(a_t|s_t, \omega_j)$, and transitional skills $\pi(a_t|s_t, z_{i,j,k})$. Specifically, we discover skills by maximizing the following mutual information

$$
\begin{aligned}
L(\theta) \quad &\triangleq \quad \mathrm{I}(S^z; \Omega_i) + \mathrm{I}(S^z; \Omega_j) \\
&= \quad \mathbb{E}_{\omega_i \sim p(\omega), s^z \sim \pi(a_t|s_t, z_{i,j,k})} \left[ \log p(\omega_i|s^z) - \log p(\omega_i) \right] \\
&\quad + \quad \mathbb{E}_{\omega_j \sim p(\omega), s^z \sim \pi(a_t|s_t, z_{i,j,k})} \left[ \log p(\omega_j|s^z) - \log p(\omega_j) \right],
\end{aligned}
\tag{3}
$$

where $S^z$ represents one of states $S_{i,j,k}^t$ at time stamp $t$, $1 \leq k \leq K-1$, in Figure 1. It is worth being noted that when the starting primitive skill is same with the ending primitive skill, no transitional skill needs to be discovered. In such case, maximizing $L(\theta)$ in (3) is equivalent to maximizing $\mathrm{I}(S^z; \Omega_i)$, which is identical to learning only primitive skills Gregor et al. (2016); Eysenbach et al. (2018).

The challege is how to solve such problem associated with two primitive skills and acquire the intrinsic reward. Our solution is to use a compensation for individual mutual information so that two components become separable.

It is observed that, when the transitional skill $\pi(a_t|s_t, z_{i,j,k})$ transits from the starting primitive skill $\pi(a_t|s_t, \omega_i)$ to the ending primitive skill $\pi(a_t|s_t, \omega_j)$, the mutual information $\mathrm{I}(S^z; \Omega_i)$ decreases progressively while the mutual information $\mathrm{I}(S^z; \Omega_j)$ increases progressively.

In order to compensate the mutual information, we define $f_{\omega_i, z_{i,j,k}}^d$ as the divergence of the latent variables $\omega_i$ and $z_{i,j,k}$, which indicates the deviation of $z_{i,j,k}$ from $\omega_i$. Similarly, we define $f_{z_{i,j,k}, \omega_j}^d$ as the divergence of the latent variables $z_{i,j,k}$ and $\omega_j$, which indicates the deviation of $\omega_j$ from $z_{i,j,k}$. The divergence defined here can be measured in a couple of ways [3]. Therefore, the objective function to be maximized becomes

$$
\begin{aligned}
L(\theta) \quad &\simeq \quad \mathrm{I}(S^z; \Omega_i) + \mathrm{I}(S^z; \Omega_j) + f_{\omega_i, z_{i,j,k}}^d + f_{z_{i,j,k}, \omega_j}^d \\
&= \quad \underbrace{\mathbb{E}_{\omega_i \sim p(\omega), s^z \sim \pi(a_t|s_t, z_{i,j,k})} \left[ \log p(\omega_i|s^z) - \log p(\omega_i) \right] + f_{\omega_i, z_{i,j,k}}^d}_{L_1(\theta)} \\
&\quad + \underbrace{\mathbb{E}_{\omega_j \sim p(\omega), s^z \sim \pi(a_t|s_t, z_{i,j,k})} \left[ \log p(\omega_j|s^z) - \log p(\omega_j) \right] + f_{z_{i,j,k}, \omega_j}^d}_{L_2(\theta)}.
\end{aligned}
\tag{4}
$$

The sufficient condition of maximizing the objecive function in (4) is to simultaneously maximize $L_1(\theta)$ and $L_2(\theta)$. It is further observed that, when the transitional skill $\pi(a_t|s_t, z_{i,j,k})$ transits from

---

[3]For details, please see it in Section 4: Implementation.

$\pi(a_t|s_t, \omega_i)$ to $\pi(a_t|s_t, \omega_j)$, the divergence $f^d_{\omega_i, z_{i,j,k}}$ increases progressively while the divergence $f^d_{z_{i,j,k}, \omega_j}$ decreases progressively. Thus, the first component $L_1(\theta)$ that has a decreasing mutual information and an increasing divergence, is symmetric to the second component $L_2(\theta)$ that has an increasing mutual information and a decreasing divergence. The sole difference lies in the transition from $\pi(a_t|s_t, \omega_i)$ to $\pi(a_t|s_t, \omega_j)$ or from $\pi(a_t|s_t, \omega_j)$ to $\pi(a_t|s_t, \omega_i)$. So maximizing $L_1(\theta)$ is equivalent to maximizing $L_2(\theta)$.

Consequently, maximizing the overall objective function $L(\theta)$ is equivalent to maximizing $L_1(\theta)$ or $L_2(\theta)$. In such case, we have

$$
\begin{aligned}
\arg\max L(\theta) \quad &\approx \quad \arg\max L_1(\theta) \\
&= \quad \arg\max \mathbb{E}_{\omega_i \sim p(\omega), s^z \sim \pi(a_t|s_t, z_{i,j,k})} \left[ \log p(\omega_i|s^z) - \log p(\omega_i) + f^d_{\omega_i, z_{i,j,k}} \right] (5)
\end{aligned}
$$

Because it is difficult to find $p(\omega_i|s^z)$, it is common to approximate $p(\omega_i|s^z)$ with a learned discriminator $q_\phi(\omega_i|s^z)$. According to Jensen's Inequality, we know that replacing $p(\omega_i|s^z)$ with $q_\phi(\omega_i|s^z)$ gives us a variational lower bound $G(\theta, \phi)$ of $L(\theta)$. So we have

$$
\arg\max L(\theta) \Longleftrightarrow \arg\max \underbrace{\mathbb{E}_{\omega_i \sim p(\omega), s^z \sim \pi(a_t|s_t, z_{i,j,k})} \left[ \log q_\phi(\omega_i|s^z) - \log p(\omega_i) + f^d_{\omega_i, z_{i,j,k}} \right]}_{\triangleq G(\theta, \phi)}.
$$

$$(6)$$

## 4 IMPLEMENTATION

### 4.1 HINDSIGHT AND ONE-HOT ENCODING

In this section, we discuss about the implementation of learning transitional skills, where there are a couple of issues with the usage of the latent variable corresponding to transitional skills.

**Problem 1**: Along with the growth of the number of primitive skills $N$ and transitional skills $K - 1$, we have a high training complexity. In our approach, we calculate the conditional probability $q_\phi(\omega_i|s^z)$ with a low efficiency because $q_\phi(\omega_i|s^z)$ can just keep the consistency of $z_{i,j,k}$ with $\omega_i$ but ignore the relation with $\omega_j$.

To enhance the efficiency, we utilize the *hindsight experience reply mechanism* to allow sample-based learning from the sparse reward.

**Problem 2**: The categorical distribution of the latent variable $\omega_i$, $1 \le i \le N$, suffers from a dilemma when the discriminator $q_\phi(\omega_i|s^z)$ judges the states.

Our solution is to use one-hot encoding for the latent variable $\omega_i$, $1 \le i \le N$, corresponding to primitive skills. In such case, the latent variable $z_{i,j,k}$ corresponding to transitional skills has a different expression and further analysis is given in Appendix B.

In the implementation, we change $q_\phi(\omega_i|s^z)$ in (6) to the following conditional probability distribution:

$$
q_\phi(\tilde{\Omega}|s^z) = [q_\phi(\omega_0|s^z), q_\phi(\omega_1|s^z), ..., q_\phi(\omega_{N-1}|s^z)]^{\mathrm{T}}. \tag{7}
$$

Correspondingly, the divergence of $f^d_{i,j,k}$ is organized as the following $f^d_{i,k}$,

$$
f^d_{i,k} = [f^d_{i,1,k}, f^d_{i,2,k}, ..., f^d_{i,N,k}]^{\mathrm{T}}, \tag{8}
$$

where $k \ne i$. Consequently, the objective is to maximize the following $G(\theta, \phi)$,

$$
G(\theta, \phi) = \frac{1}{N} \cdot \mathrm{E}_{\omega_i \sim p(\omega), s^z \sim \pi(z_{i,k})} (\| \log q_\phi(\tilde{\Omega}|s^z) + \alpha f^d_{i,k} \|_2), \tag{9}
$$

where $z_{i,k} \sim p_z(z|\omega_i) = \{z_{i,k}|z_{i,k} = z_{i,j,k}, 0 \le j \le N-1, z_{i,j,k} \in p(z_{i,j}|\omega_i, \omega_j)\}$, $\alpha$ is a hyperparameter, and we ignore the term $p(\omega_i)$ by taking a uniform distribution for $\omega_i$.

---

**Algorithm 1** Learning Transitional Skills (LTS)

1: **while** NOT converged **do**
2:     Sample $\omega_i \sim p(\omega)$
3:     Sample a skill $z \sim p_z(z|\omega_i)$ and an initial state $s_0 \sim p_0(s)$
4:     **for** $t \leftarrow 1$ **to** steps per episode **do**
5:         Sample an action $a_t \sim \pi_\theta(a_t|s_t, z)$;
6:         Interact with the environment: $s_{t+1} \sim p(s_{t+1}|s_t, a_t)$;
7:         Compute $D_t = \frac{1}{N}\|q_\phi(\tilde{\Omega}|s_{t+1}) + \alpha f_{i,k}^d\|_2$ with the discriminator $(\phi)$;
8:         Set the reward for current skill: $r_t = D_t$.
9:         By using SAC, update the policy $(\theta)$ to maximize the discounted return $R = \sum_{t=0}^{\infty} \gamma^t r_t$;
10:        Update the discriminator $(\phi)$ to maximize $D_t$ with SGD.
11:    **end for**
12: **end while**

---

## 4.2 ALGORITHM

We summarize our LTS method in Algorithm 1. At each roll-out, given the latent variable of primitive skill $\omega_i$, we sample a transitional skill $z$ from a fixed distribution $p_z(z|\omega_i)$. After the agent interacts with the environment at time step $t$, the discriminator finds the discriminability by

$$D_t = \frac{1}{N}\|q_\phi(\tilde{\Omega}|s_{t+1}^z) + \alpha f_{i,k}^d\|_2, \tag{10}$$

where $s_{t+1}^z$ denotes the states at time step t. As mentioned above, we encode primitive skills using one-hot encoding. We constrain $\sum_j z_{i,j,k} = 1$, $0 \leq z_{i,j,k} \leq 1$, so we simplify the divergence as $f_{i,k}^d = \text{Array}(z_{i,j,k})$, where $\text{Array}(\cdot)$ means convert one-hot encoding into an array. (See example in Appendix B.1 for more details (16, 17)).

In addition, we adopt soft actor-critic (SAC) algorithm to train our policy, adding the regularization $\mathbb{E}_{i,j}[\text{H}[A|S, Z_{i,j}]]$ to maximize the policy's entropy over actions given states and skills.

## 5 RELATED WORK

Real-world tasks often require diverse behaviors. Wang et al. (2017) notes that building versatile embodied agents capable of performing a wide and diverse set of behaviors is one of the long-standing challenges of AI. And learning continuous control of diverse behaviors in locomotion (Merel et al. (2017); Heess et al. (2017); Peng et al. (2017)) and robotic manipulation (Ghosh et al. (2018); Gu et al. (2017)) is an active research area. In this scenario, although some complex tasks can be solved through extensive reward engineering, undesired behaviors often emerge because of the sparse nature of reward (Riedmiller et al. (2018)). Moreover, training complex skills from scratch is not computationally practical. These issues can be addressed by use of intrinsic motivation (Barto (2013); Chentanez et al. (2005); Singh et al. (2010)), which is a reward-free learning method. Historically, the intrinsic motivation comes from the tendency of organisms to play and explore their environment without any reward (Ryan & Deci (2000), White (1959)).

Another line of work that is conceptually close to our method copes with information theories that are used to drive the agent's exploration. The information gain is a reward based on the reduction of uncertainty on environment's dynamics (Little & Sommer (2013); Oudeyer & Kaplan (2007)), which can also be assimilated to learning progress (Frank et al. (2013); Oudeyer & Kaplan (2007)). This can push agents into unknown areas on the one hand, and prevent them from being attracted to random areas on the other.

Recent work has also applied information theory for skill discovery. VIC (Gregor et al. (2016)) is an optional discovery technique by optimizing a variational lower bound on the mutual information between the context and the final state in a trajectory, conditioned on the initial state. Furthermore, DIAYN (Eysenbach et al. (2018)) maximizes the mutual information between states and skill to achieve diversity and shows the interest as a pre-training for hierarchical reinforcement learning or as an initialization for learning a task. While discriminative embedding reward networks (DISCERN) (Warde-Farley et al. (2018) aim to simultaneously learn a goal-conditioned policy and a

goal achievement reward function by maximizing the mutual information between the goal state and the achieved state. Let us notice that the skill space here is discrete, with just one or multiple policies. However, we considered the relationship between different skills during the training process and finally formed a continuous skill space, likely because of inducing a novel latent variable for transitional skills.

In addition, it is important to point out that our skills are transitional with an intrinsically driven approach, which is very different from numerous previous works. While Sharma et al. (2019) discovers predictable behaviors to let the single skill more predictable, it need an external model-predictive-control (MPC) paradigm (Garcia et al. (1989)) to connect skills. Peng et al. (2019) learns reusable motor primitives that can be composed to produce a continuous spectrum of skills. To bridge the gap between skills, Lee et al. (2018) propose a transition policy to get a new smooth skill. In comparison, our method captures intrinsic transition, which is independent from external tasks, and could eliminate the extra fine-tuning process.

## 6 EXPERIMENTS

In our experiments, we aim to demonstrate the effectiveness of our approach for learning primitive and transitional skills. We evaluate LTS and compare it to prior works.

### 6.1 DIVERSITY AND TRANSITIONAL SKILLS

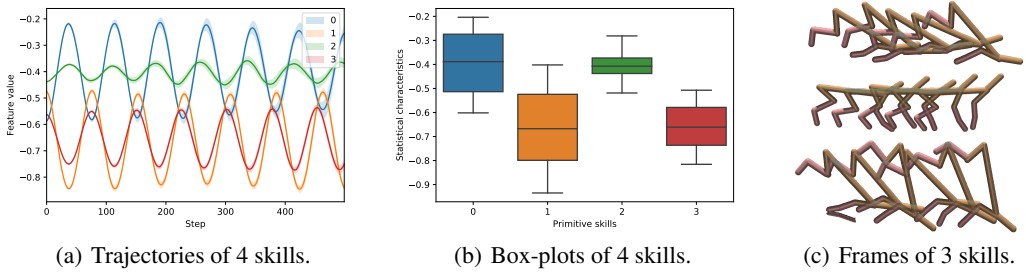

(a) Trajectories of 4 skills.     (b) Box-plots of 4 skills.     (c) Frames of 3 skills.

Figure 2: Diverse primitive skills ($\omega_0$, $\omega_1$, $\omega_2$, $\omega_3$) in MountainCar (a, b) and HalfCheetah (c). (a) shows the changes of the feature values of 4 diverse skills. (b) shows the corresponding statistical characteristics of 4 skills in (a), where a single box-plot denotes a skill. (c) shows 3 diverse skills of the half-cheetah.

In this section, we provide visualizations and quantitative analysis for our LTS method. The tasks of CartPole-v0, MountainCar-v0, Pendulum-v0 and HalfCheetah-v3 are based on OpenAI gym [4].

To evaluate different skills in different environment, we extract features from the observation of the agent, e.g. in the MountainCar environment, where we use the variance of the car's altitude as the feature. For more details on the features in other environments, please refer to Appendix C.

Figure 2 illustrates the discovered primitive skills in the MountainCar and Halfcheetah environment. As shown in Figure 2(a), all four skills move in an periodic manner. Corresponding to all 4 skills in Figure 2(a), the statistical values of features are shown in Figure 2(b) using Box-plot. It is observed that, these four skills have different movement patterns so that these skills are easy to be distinguished, so as the skills shown in Figure 2(c). Moreover, we consider a different number of primitive skills and different environments as in Appendix E, from which it is observed that all primitive skills have an evident difference in feature statistics and are easy to be distinguished.

**Transition.** Furthermore, we use the transitional skill $z_{0,1,k}$ to show the performance of transition from one primitive skill $\omega_0 = [1, 0, 0, 0])$ to another $\omega_1 = [0, 1, 0, 0])$, where the number of transitional skills is set as 9.

---

[4]http://gym.openai.com/

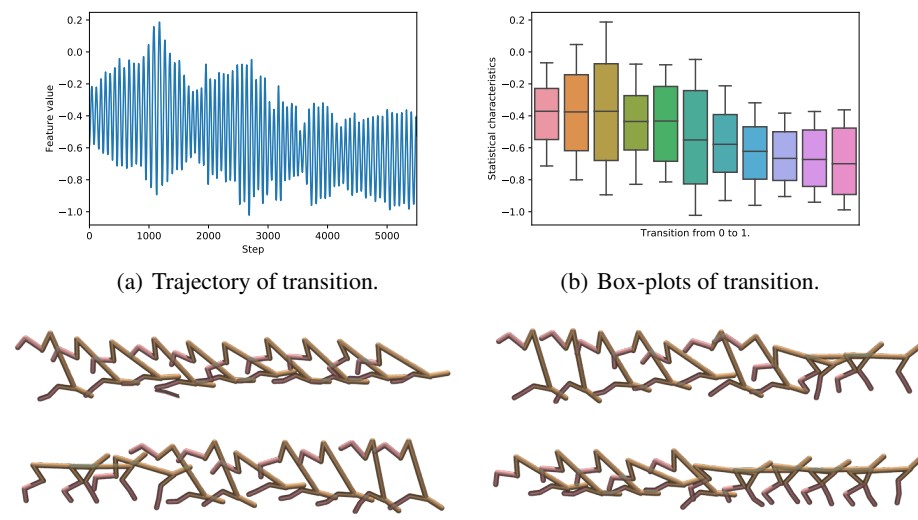

(a) Trajectory of transition.

(b) Box-plots of transition.

(c) Illustration of 4 transitional skills.

Figure 3: Transition in Mountain-Car (a,b) and HalfCheetah (c). (a) shows the transition of the trajectory, where 9 transitional skills are uniformly distributed along the horizontal axis from Step $0$ to $5500$. (b) shows the transition of feature statistics, where the features of two primitive skills and 9 transitional skills are included. (c) shows the sampled frames of the transition trajectory, where $K - 1 = 9$.

In Figure 3(a) and 3(b), it is observed that the primitive skill $\omega_0$ smoothly changes to $\omega_1$ via 9 transitional skills. More specifically, there exists a slight increment on the amplitude of features in the first three skills, which is followed by consecutive declines until the ending primitive skill $\omega_1$ is discovered. In Figure 3(c), we can see the clear transition process between two primitive skills. More experiments on skill transition are given in Appendix F. This demonstrates the effectiveness of our LTS method on discovering transitional skills and accomplishing the successful transition between two primitive skills.

## 6.2 COMPARISON WITH DIAYN

In this subsection, we compare our LTS method with the state-of-the-art DIAYN (Eysenbach et al. (2018)) in terms of learning diverse primitive skills and transitional skills. Experimental results show that LTS achieves an approximate performance on diversity of primitive skills, and a much better performance on skill transition.

**Diversity:** Different skills have different means or variances from their own trajectories. In this subsection, we use (1) *variance of means* and (2) *mean of variances* of the features as metrics to evaluate the diversity of learned primitive skills using LTS and DIAYN.

Figure 4(a) shows the variance of means while Figure 4(b) shows the mean of variances, where the black perpendicular line represents the range of values from a single experiment and we collect all trajectories of various primitive skills. In Figure 4(a), the height of this line depicts the range of variance of means by considering all trajectories of primitive skills. As shown, LTS obtains a lower variance of means in compare to DIAYN because the learned transitional skills from LTS affect the variance of learned primitive skills.

The lower variance of means does not degrade the diversity of learned primitive skills, which can be observed from Figure 4(b). Although there is a large difference between experiments with random seeds, LTS has generally a similar mean of variance as DIAYN, indicating that our method performs similar with the baseline in terms of learning diverse primitive skills.

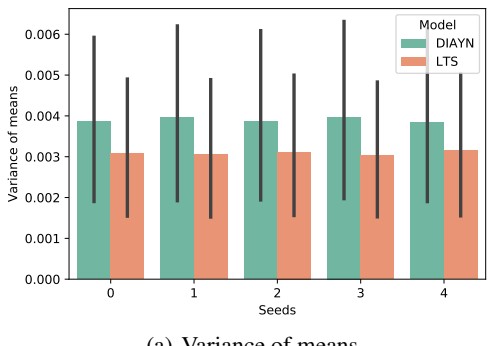
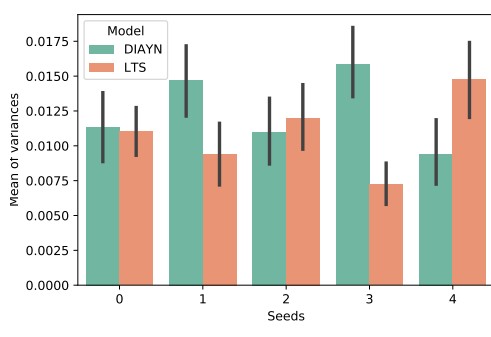

(a) Variance of means.

(b) Mean of Variances.

Figure 4: Comparison of diversity of primitive skills between LTS and DIAYN.

**Transition:** We also conduct experiments to evaluate the transitional skills of LTS and DI-AYN, where we use identical encoding scheme for both and the number of transitional skills is set as 9.

Figure 5 shows the all statistical characteristics of 11 skills that including two primitive skills, where the value denotes the mean of features like the middle bar in Figure 3(b). It is observed that LTS experiences a smooth transition state from one primitive skill to another while the transitional states in DIAYN suffer from one

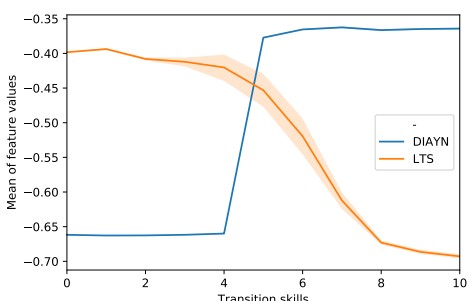

Figure 5: Comparison of transitional skills.

sharp-rising phase and two steady phases indicating a rigid changes of transition states. The steady phase indicates that changing $z$ will not change states very much, and the sharp change indicates that a little fluctuate of $z$ will cause a huge change of states, which is not reasonable in practice. Because of the huge gap, controlling the skills learned by DIAYN changing from $\omega_0$ to $\omega_1$ will not push the agent standing in the state space of $\omega_2$. On the contrary, we control the transitional skills in LTS changing from $\omega_0$ to $\omega_1$ as in Figure 2(b), and the agent reaches the target states.

More studies including real statistical characteristics of different transitional skills are conducted and the results are reported in Appendix G.

### 6.3 GENERALIZATION

Our LTS method suffers from the high training complexity to learn transitional skills. A reasonable approach to tackle this problem is to use a fixed number of transitional skills. In this experiment, we set this number as 3, indicating that the nonzero element of $z_{i,j,k}$ is from the set $\{0.25, 0.50, 0.75\}$ in the training phase.

Figure 6 shows the transitional skills, where the training phase considers 3 transitional skills and adopting 3 and 50 transitional skills to evaluate the generalization of our LTS method. It is observed that the transitional skills in blue (traning and testing both with 3 transitional skills) suffers from severe declines, possibly leading

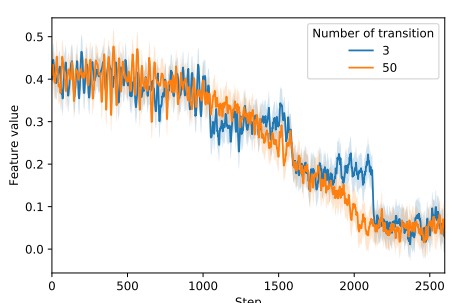

Figure 6: Transitional skills with related to 2 different number of them.

to the failure of the transition process in practice. Fortunately, the transitional skills in red (training with 3 transitional skills while testing with 50) go through a steady and smooth process from one

Table 1: Success rate(%) for practical tasks, where our method is compared with the baselines with or without hierarchical framework. Our transitional skills outperforms the baselines in all cases, demonstrating the effectiveness of the learning transitional skills for down-stream tasks. Here we adopt spectral clustering to cluster the trajectories. When the label of ending segment of the trajectory keeps the same label with that generated the target skills solely, the transition is regarded as a success; otherwise a failure transition. The target skills are regard as the goal of our tasks, which are generated by sample primitive skills or weighting the actions of different primitives.

| Policy | DIAYN | | | LTS | | |
| --- | --- | --- | --- | --- | --- | --- |
| | random | smooth | master | random | smooth | master |
| MountainCar | 0 | 0 | 20.8 | 0.2 | 50.2 | 100 |
| Pendulum | 0 | 14.2 | 100 | 1 | 100 | 100 |
| CartPole | 1.6 | 10 | 36.8 | 0 | 25.8 | 48 |
| HalfCheetah | 0 | 29.2 | 44.7 | 0 | 43.0 | 78.6 |

primitive skill to another. This experiment demonstrates that the strong generalization of LTS can improve the efficiency of the training. These means that even we use a small number of transitional skills in training, we could sample the transitional skills with a higher density and achieve the success of the skill transition, by which we could further improve the efficiency of training model.

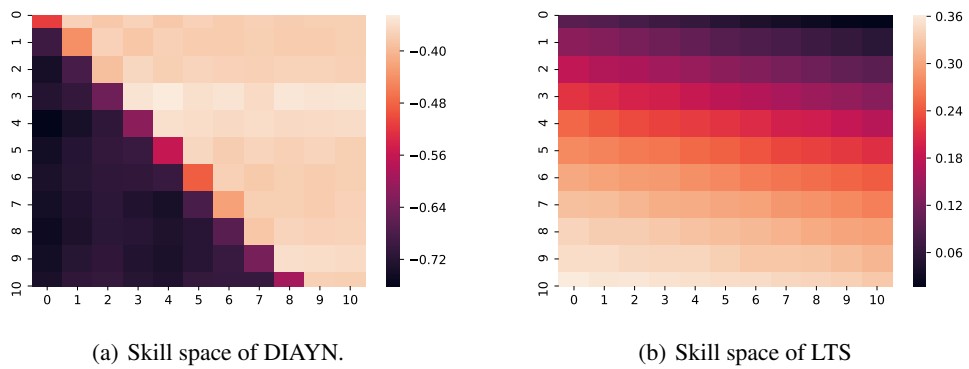

(a) Skill space of DIAYN.    (b) Skill space of LTS

Figure 7: Skill space of DIAYN and LTS.

## 6.4 EQUIP THE AGENT WITH MORE TRANSITIONAL SKILLS

Figure 7 shows much more skills and corresponding transitions between them, where each small square denotes a different skill, the number of primitive skills is 2 and the number of transitional skills is 98. Two primitive skills $[1, 0, 0, 0]$ and $[0, 1, 0, 0]$ locate at the lower left and upper right squares.

It is observed that our LTS method is able to accomplish a smooth transition between two arbitrary skills, whatever primitive skills or transitional skills. However, DIAYN suffers from a rigid transition between two skills, resulting in a possible failure in practical transition. Furthermore, these results provide a deep insight that LTS has the ability to learn a larger *continuous* skill space. The agent equipped with such numerous skills is expected to become much more powerful.

## 6.5 TRANSITION WITH HIERARCHICAL FRAMEWORK

In this subsection, we conduct experiments to answer the question: how do transitional skills yield benefit to the downstream tasks? To evaluate the effect of the transition policies, we take as benchmark hierarchical control for primitives learned by DIAYN. Moreover, we conduct hierarchical framework to weight (or choose) the action modeled by different primitive skills.

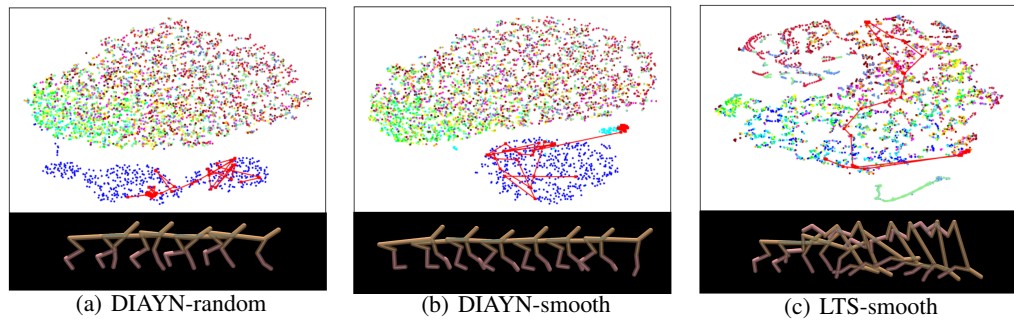

| (a) DIAYN-random | (b) DIAYN-smooth | (c) LTS-smooth |

Figure 8: Visualization of transition trajectories of (a) *DIAYN-random*, (b) *DIAYN-smooth* and (c) *LTS-smooth*. Top row contains the states generated by different skills noted by the dots and the red lines represent the agent's sampled trajectories, both of which are mapped to a two-dimensional space by t-SNE. Bottom contains the frames of transition trajectories.

For primitive skills modeled by DIAYN, we consider the following approaches:1) *DIAYN-random*: Transition policy with random weights to affect actions; 2) *DIAYN-smooth*: Transition policy with smoothly changed weights (artificially designed) to affect actions; 3) *DIAYN-master*: Transition policy with optimal weights learned by meta-policy.

For transitional skills modeled by LTS, we consider the following approaches: 1) *LTS-random*: Randomly choose transitional skills between two primitive skills; 2) *LTS-smooth*: Choosing continously changed transitional skills $z_{i,j,k}$ ( $1 \leq k \leq K-1$); 3) *LTS-master*: Transfer policy with optimal weights learned by meta-policy.

Table 1 shows the performance. It is observed that we choose skills randomly or weighted actions modeled by primitive skills, the transition process is more likely to fail. Even with an extensively forecast through the hierarchical framework, *DIAYN-master* is possible to fail. We believe this is because the primitive skills have a large gap and therefore, one skill are unable to associate with another. Even with meta-policy, it is also hard to transit between primitive skills. In contrast, *LTS-smooth* has a better performance than *DIAYN-master* without any meta-policy because of abundant transitional skills. Moreover, *LTS-master* could select most appropriate transitional skills to fill in the gap between two arbitrary skills.

## 6.6 VISUALIZING TRANSITION TRAJECTORIES

Figure 8 shows the two-dimensional t-SNE embedding of the agent's states generated by different skills. Different colors represent different skills (including primitive skills and transitional skills). The red lines represent the agents' transition trajectories. In Figure 8(a) and 8(b), we can see a clear boundary between two kinds of skills, which is consistent with the boundary of Figure 7.

Starting from the initiatial state, the agent tries to transit to another state. But both of agents with DIAYN-random-policy and DIAYN-smooth-policy fail for transiton. The state space of different skills in Figure 8(c) seems to be more diverse, while it is a pseudomorphism. Because the space is a two-dimensional mapping, the difference between Figure 8(a) and 8(b) only discloses that different skills do exist a conclusive gap between two kinds of skills, but this uncovers that the learned transitional skills are also distinguishable. We can find in Figure 8(c) also exits many intersections which are the crucial transition states between two different skills. In the t-SNE space, distant dots with same color are also connective, so there exits many long line. And the intersections existing two endpoints must go through a transition between two skills. Particularly, the trajectory of the agent in Figure 8(c) shows the transition between different 11 skills.

## 7 CONCLUSION

In this paper, we introduce a novel LTS method to learn transitional skills without extrinsic reward by using two kinds of latent varibles to depict different skills. As a result, LTS can discover both

primitive skills and transitional skills. Also, LTS achieves a great success in the smooth transition from one primitive skill to another and exhibits its potential in learning a large continuous skill space. Extensive experiments demonstrate the effectiveness of our LTS in the discovery of diverse skills and the smooth transition between skills even for complex down-stream tasks.

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

**Appendices**

## A  METHODOLOGY

### A.1  DERIVATION OF THE VARIATION BOUND ON MUTUAL INFORMATION

Here we derive the variational bound:

$$
\begin{aligned}
\mathrm{E}_{\omega_i \sim p(\omega)} \left[\log p(\omega_i|s_t)\right] &= \mathrm{E}_{\omega_i \sim p(\omega)} \left[\log q_\phi(\omega_i|s_t)) + \alpha KL(p(\omega_i|s_t)|q_\phi(\omega_i|s_t)\right] \\
&\geq \mathrm{E}_{\omega_i \sim p(\omega)} \left[\log q_\phi(\omega_i|s_t)\right]
\end{aligned}
\tag{11}
$$

## B  IMPLEMENTATION

### B.1  HINDSIGHT AND ONE-HOT ENCODING

In our approach, we model the conditional probability given by the discriminator $q_\phi(\omega_i|s_t)$ w.r.t. the divergence (or distance) between transitional skill $z_{i,j,k}$ and specific primitive skill $\omega_i$. Although (4) constrains two mutual information terms, but (**??**) is adopted to train our model, which constrains only one term. These efficiency is relatively low. For example, if we want to simultaneously constrain the probability $q_\phi(\omega_{i_1}|s_t)$ and $q_\phi(\omega_{i_2}|s_t)$ in terms of the same transition state $s_t$ generated by the same skill $z$ ($z = z_{i1,j1,k1} = z_{i2,j2,k2}$) but two different primitive skills ($\omega_{i1}$ and $\omega_{i2}$) in the training stage, we must wait for the next time step to sample the same $z$ and a different $\omega$ in the experience. The efficiency is relatively low because that $q_\phi(\omega_i|s_t)$ just constrain the consistency of $z_{i,j,k}$ and $\omega_i$ which ignores the consistency to other primitive skills. So we utilize the hindsight experience reply mechanism to allow sample-efficient learning form sparse rewards. We calculate a distribution of the conditional probability given by the discriminator instead of a single value. So we change $q_\phi(\omega_i|s_t)$ to the conditional probability distribution:

$$
q_\phi(\Omega, s_t) = [q_\phi(\omega_0|s_t), q_\phi(\omega_1|s_t), ..., q_\phi(\omega_{N-1}|s_t)]^{\mathrm{T}}.
\tag{12}
$$

By doing this, we could simultaneously constrain the similarity probability distribution given by the discriminator with related to all primitive skills.

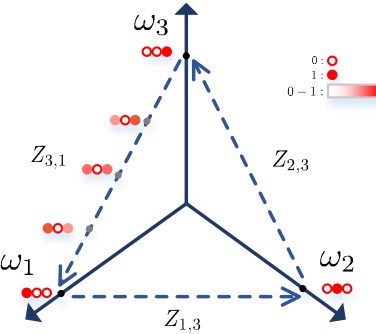

Figure 9: The space of primitive skills and transitional skills, where we set $N = 3$, $K - 1 = 3$.

Moreover, we change the criterion of $f_{i,j,k}^d$ into $f_{i,k}^d$:

$$
f_{i,k}^d = [f_{i,1,k}^d, f_{i,2,k}^d, ..., f_{i,N-1,k}^d]^{\mathrm{T}}.
\tag{13}
$$

On the other hand, for categorical encoded primitive skills, when $\omega_i - \omega_j \neq \pm 1$ ($i \neq j$), the discriminator $q_\phi$ will face a dilemma: the intersection of transition states and primitive states are not empty, i.e. $S_{i,j}^T \cap S^P \neq \oslash$, leading to a conflict between diversity and transition. Here we give an example: considering transferring two skills from $\omega_i$ to $\omega_j$ ($\omega_i \neq \omega_j$), the former categorical encoding will cause extra consumption: if $\omega_i - \omega_j \neq \pm 1$, primitive states with related to primitive skills $\{\omega|\omega \in [\omega_i, \omega_j] \text{ or } [\omega_j, \omega_i]\}$ will occur in the transition states. For three primitive skills $\omega_{i_1}$,

$\omega_{i_2}$ and $\omega_{i_3}$, given the optimal discriminator $q_\phi^*$, there should be $q_\phi^*(\omega_{i_3}|s_t') = 0$ ($s_t' \sim \pi(\omega_{i_1})$ and $\omega_{i_1} \neq \omega_{i_3}$) and $q_\phi^*(\omega_{i_3}|s_t'') = 0$ ($s_t'' \sim \pi(\omega_{i_2})$ and $\omega_{i_2} \neq \omega_{i_3}$). While if $\omega_{i_1} < \omega_{i_3} < \omega_{i_2}$, $q_\phi^*(\omega_3|s_t') = \log p(\omega_{i_1}) - f^d(\omega_{i_1}) + c$ and $q_\phi^*(\omega_3|s_t'') = \log p(\omega_{i_2}) - f^d(\omega_{i_2}) + c$ [5], which is in contrast to the conditional probability of 1 ($\omega_{i_3}$ is also a primitive skill). So, we encode $\omega \sim p(\omega)$ with one-hot way:

$$\begin{aligned} \omega_0 &= [1, 0, 0, ..., 0]; \\ \omega_1 &= [0, 1, 0, ..., 0]; \\ &... \\ \omega_{N-1} &= [0, 0, 0, ..., 1]. \end{aligned} \tag{14}$$

And we denote primitive skills and transitional skills as a set $Z_{i,j}^+$:

$$\begin{aligned} Z_{i,j}^+ &= [\omega_i, z_{i,j,1}, ..., z_{i,j,k}, ..., z_{i,j,K-1}, \omega_j]^{\mathrm{T}} \\ &= \begin{bmatrix} 0 & ... & 1 & ... & 0 & ... & 0 \\ 0 & ... & 1 - \frac{1}{K} & ... & \frac{1}{K} & ... & 0 \\ .... & & & & & & \\ 0 & ... & 1 - \frac{k}{K} & ... & \frac{k}{K} & ... & 0 \\ ... & & & & & & \\ 0 & ... & 0 & ... & 1 & ... & 0 \end{bmatrix}_{(K+1) \times N} \end{aligned}, \tag{15}$$

where the value of $i$-th and $j$-th column keeps decreasing and increasing respectively. Other column always keep 0, which could constrain the incoherence between transition states and other primitive skills. Without causing any misunderstanding, following $z_{i,j,k}$ all comes from $Z_{i,j}^+$. For transition, we assure that the change of $z_{i,j,k}$ only happens on the corresponding dimension, which overcomes the conflict caused by categorical encoding. As show in Fig.9, all transition skills in $Z_{3,1}$ and primitive skills $\omega_3$ are orthogonal. The transition only reflects on the plane defined by the corresponding primitive skills. Although these will induce a problem of the generation issue, but this could also learning continuous transitional skills as shown in Figure 6 and 7. In fact, there is more than one transitional path, which can be a directed line or any directed curve. As in Figure 7(b), we can find more than one transitional paths.

An example $Z_{i,j}^+$ shows below:

$$\begin{aligned} Z_{3,1}^+ &= [\omega_3, z_{3,1,1}, z_{3,1,2}, z_{3,1,3}, \omega_1]^{\mathrm{T}} \\ &= \begin{bmatrix} 0 & 0 & 0 & 1 & 0 \\ 0 & 0.25 & 0 & 0.75 & 0 \\ 0 & 0.5 & 0 & 0.5 & 0 \\ 0 & 0.75 & 0 & 0.25 & 0 \\ 0 & 1 & 0 & 0 & 0 \end{bmatrix}_{5 \times 4} \end{aligned}, \tag{16}$$

where $N = 4$, $K - 1 = 3$. A transition path could be $\omega_3 \rightarrow z_{3,1,1} \rightarrow z_{3,1,2} \rightarrow z_{3,1,3} \rightarrow \omega_1$. And an alternative way could be $[0, 0, 0, 1, 0] \rightarrow [0, 0.2, 0, 0.8, 0] \rightarrow [0.1, 0.3, 0.2, 0.3, 0.1] \rightarrow [0, 0.8, 0, 0.2, 0] \rightarrow [0, 1, 0, 1, 0]$. For the similairity $f_{i,k}^d$, examples are shown below (generated by $Z_{3,1}^+$ in Formula(16) ):

$$\begin{aligned} f_{3,0}^s &= [0, 0, 0, 1, 0]^{\mathrm{T}}; \\ f_{3,1}^s &= [0, 0.25, 0, 0.75, 0]^{\mathrm{T}}; \\ f_{3,2}^s &= [0, 0.5, 0, 0.5, 0]^{\mathrm{T}}; \\ f_{3,3}^s &= [0, 0.75, 0, 0.25, 0]^{\mathrm{T}}; \\ f_{3,4}^s &= [0, 1, 0, 0, 0]^{\mathrm{T}}, \end{aligned} \tag{17}$$

where $f_{i,j}^s = 1 - f_{i,j}^d$.

---

[5] $c$ is a constant.

## C   EXPERIMENTAL ENVIRONMENT

The experiments were carried out over four opened reinforcement learning environments (CartPole[6], MountainCar[7], Pendulum[8] and HalfCheetah[9].

### C.1   CARTPOLE

In this environment, a pole is attached by an un-actuated joint to a cart, which moves along a frictionless track. The system is controlled by applying a force of +1 or -1 to the cart. The pendulum starts upright, and the goal is to prevent it from falling over by increaseing and reducing the cart's velocity. The episode ends when the pole is more than 15 degrees from vertical, or the cart moves more than 2.4 units from the center.

### C.2   MOUNTAINCAR

A car is on a one-dimensional track, positioned between two "mountains". The goal is to drive up the mountain on the right; however, the car's engine is not strong enough to scale the mountain in a single pass. Therefore, the only way to succeed is to drive back and forth to build up momentum.

### C.3   PENDULUM

The inverted pendulum swingup problem is a classic problem in the control literature. The problem of the pendulum starts in a random position, and the goal is to swing it up so it stays upright.

### C.4   HALFCHEETAH

In the HalfCheetah environment, there is a two-legged robot, restricted to a vertical plane, meaning it can only run forward or backward. The agent has 17 dimensions of state space and 6 dimensions of action space.

## D   HYPERPARAMETERS

For all RL algorithm in our experiments, we use the SAC (Haarnoja et al. (2018)) as implementation framework. The hyperparameters are summed up in the Table 2 and we use ADAM (Kingma & Ba (2014)) optimizer.

## E   VISUALIZING PRIMITIVE SKILLS

In order to better visualize the distinction between skills, we did various experiments and finally determined some optimal observations as feature vector for each skill (see Table 3). Please note that the feature values adopted are aimed at visualizing the skills, and what we put into the discriminator is the all state of the agent instead the single feature value. The following experiments show that it makes sense to calculate the statistical characteristics of skills' characteristics to represent a skill. Three experiments' performance was shown in Figure 10, Figure 11 and Figure 12.

## F   VISUALIZING TRANSITION PROCESS

For Cartpole, MountainCar, Pendulum and HalfCheetah, we get 4 primitive skills, and control variant $z$ only takes 3 fixed values (0.25 0.5 0.75) during training. However, at the test phase of transition, $z$ is added every 0.1. This ways reduce the training complexity as analyzed in Section 6.3. So we can obtain 9 transition skills between any two primitive skills($z = [0, 1]$) (See Figure 13, Figure 14, and Figure 15).

---

[6]https://gym.openai.com/envs/CartPole-v0/

[7]https://gym.openai.com/envs/MountainCar-v0/

[8]https://gym.openai.com/envs/Pendulum-v0/

[9]https://gym.openai.com/envs/HalfCheetah-v2/

Table 2: Parameter setting

| Parameters | Description | Value |
|---|---|---|
| H | hidden state size | 32 for CartPole, MountainCar and Pendulum; 64 for HalfCheetah |
| layer | layer count | 3 |
| epoch | eposide size | 2*12(cpus) |
| vf_lr | value network learning rate | 1e-5 |
| dc_lr | discriminator network learning rate | 5e-4 |
| pi_lr | policy network learning rate | 3e-4 |
| max_episodes | the maximal length of episode | 250 |
| train_dc_iterv | epoch number of training discriminator network | 5 |
| train_ac_iterv | epoch number of training actor network | 1 |
| train_v_iterv | epoch number of training critic network | 1 |
| train_dc_iters | iteration number of updating discriminator network | 80 |
| train_ac_iters | iteration number of updating actor network | 50 |
| train_v_iters | iteration number of updating critic network | 1 |

Table 3: Selection of skill features.

| RL enviroment | Observations | Selected as skill feature |
|---|---|---|
| CartPole | 0: Cart Position; 1: Cart Velocity 2: Pole Angle 3: Pole Velocity at Tip | 2: Pole Angle |
| MountainCar | 0: Position 1: Velocity | 0: Position |
| Pendulum | 0: cos(Angle) 1: sin(Angle) 2: speed | 1: sin(Angle) |
| HalfCheetah | 17-dim values (including the positions and velocities) | the frames |

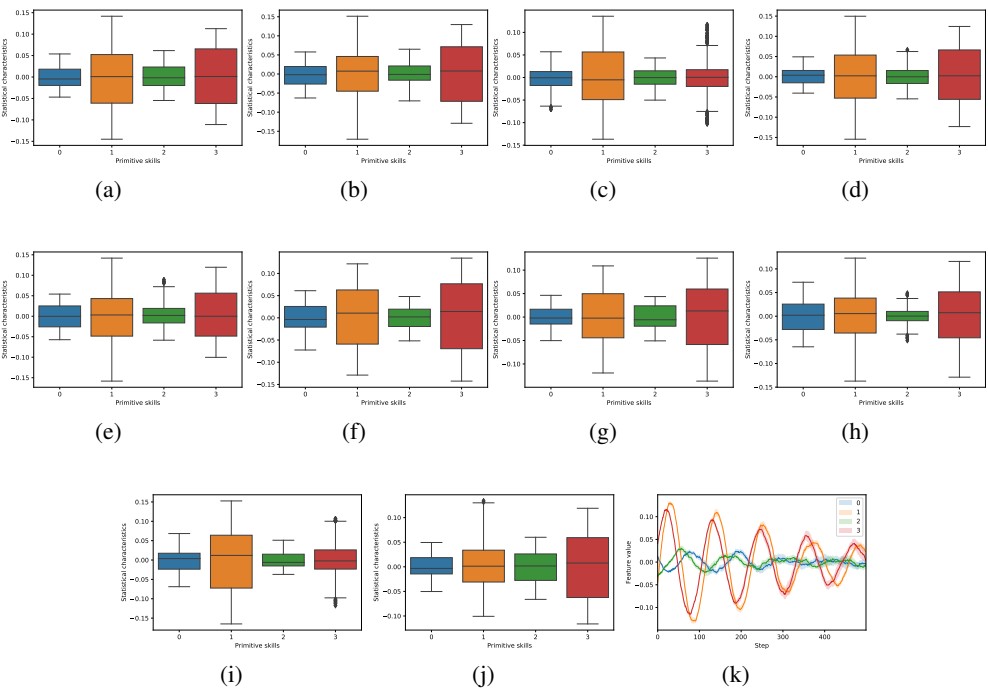

Figure 10: **Cartpole primitive skills**. (a) - (j) stand for 10 random trials with 4 primitive skills for each, and different skills are distinguished by Boxplot. (k) shows the skills in time domain for one trial.

## G   TRANSITIONAL SKILLS COMPARISON

Studies including motion trail and statistical characteristics on MountainCar was reported in Figure 16. The value of the y-axis denotes the mean of feature values (which is also the statistical characteristic). Here we do not test the transition processes between skills and we just run different transitional skills independently. There should be many isolated dots instead a transition line. Skills with two distant features will means a larger probability of failure of transition. For DIAYN's primitive skills, there exists a large gap which notes an harder transition between the two skills, comparing the smooth line of LTS's skills. Please note that there exits no faster transition instead a large gap.

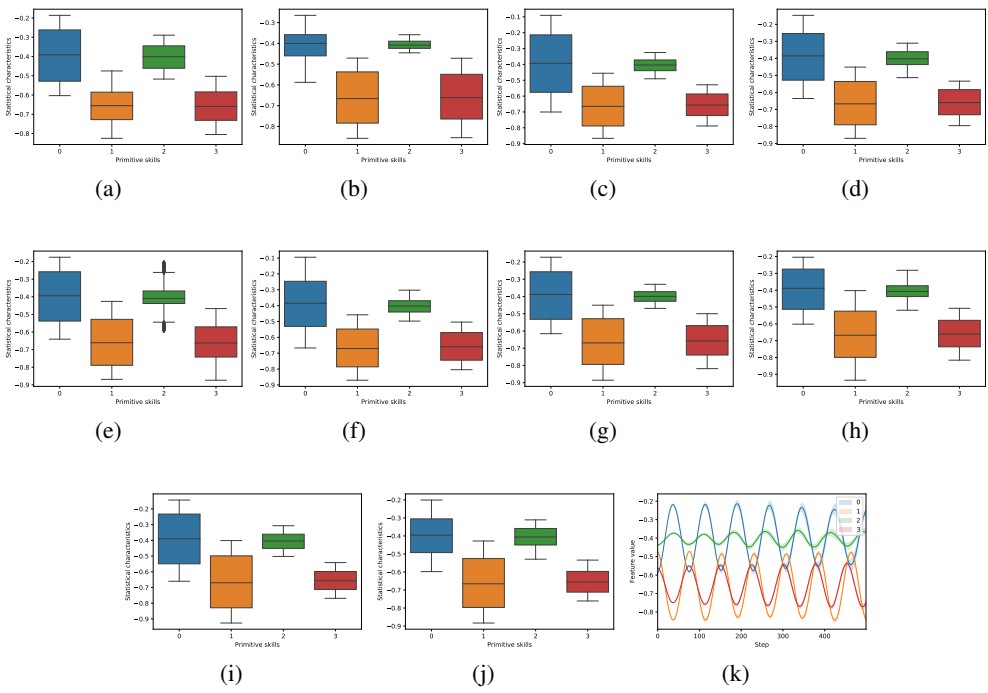

Figure 11: **Mountain Car primitive skills**. (a) - (j) stand for 10 random trials with 4 primitive skills for each, and different skills are distinguished by Boxplot. (k) shows the skills in time domain for one trial.

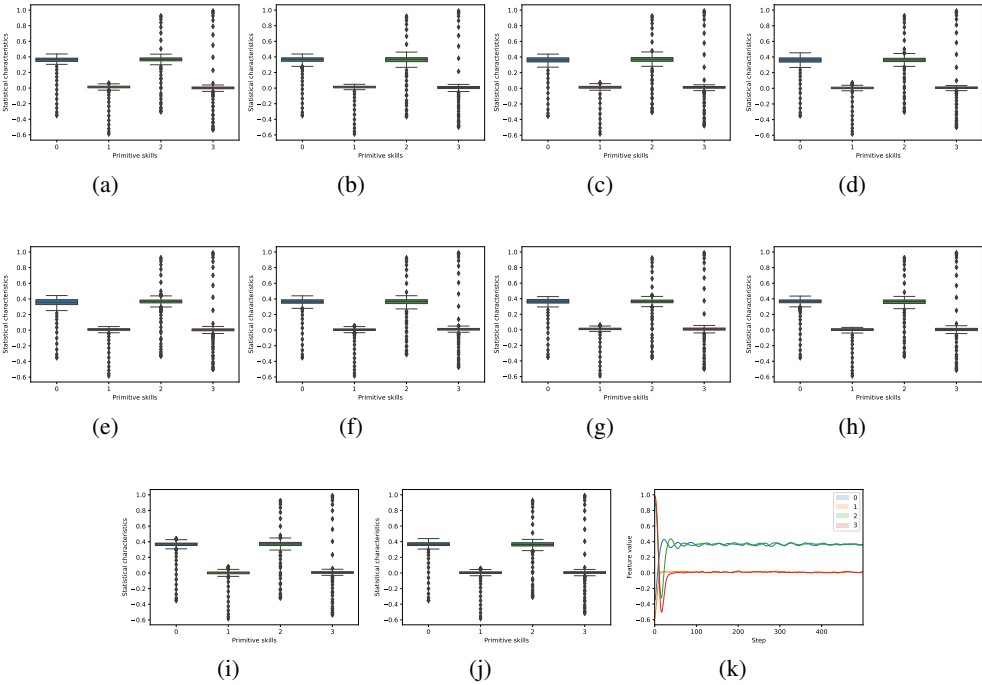

Figure 12: **Pendulum primitive skills**. (a) - (j) stand for 10 random trials with 4 primitive skills for each, and different skills are distinguished by Boxplot. (k) shows the skills in time domain for one trial.

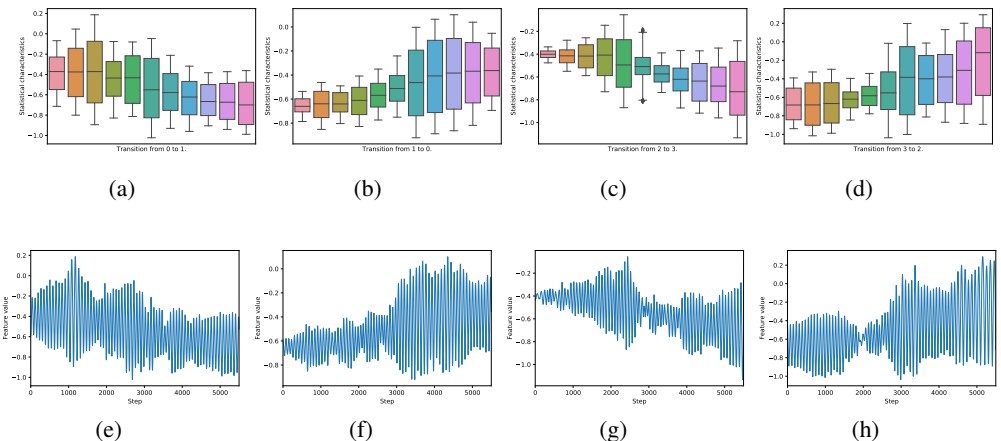

Figure 13: **Mountain Car transition process**. (a) (e), (b) (f), (c) (g), and (d) (g) are from different 4 trials respectively. Each transitional skill holds 500 steps and then transfers the final state to the next skill.

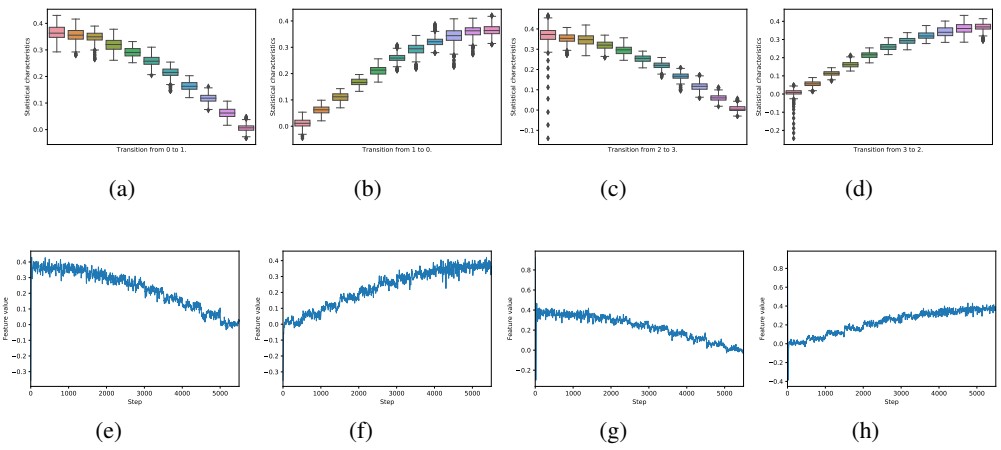

Figure 14: **Pendulum transition process**. (a) (e), (b) (f), (c) (g), and (d) (g) are from different 4 trials respectively. Each transitional skill holds 500 steps and then transfers the final state to the next skill.

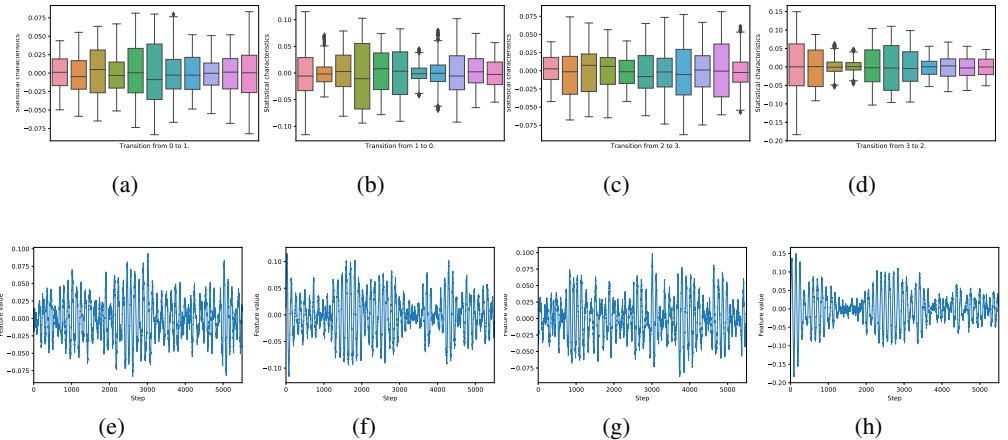

Figure 15: **Cartpole transition process**. (a) (e), (b) (f), (c) (g), and (d) (g) are from different 4 trials respectively. Each transitional skill holds 500 steps and then transfers the final state to the next skill.

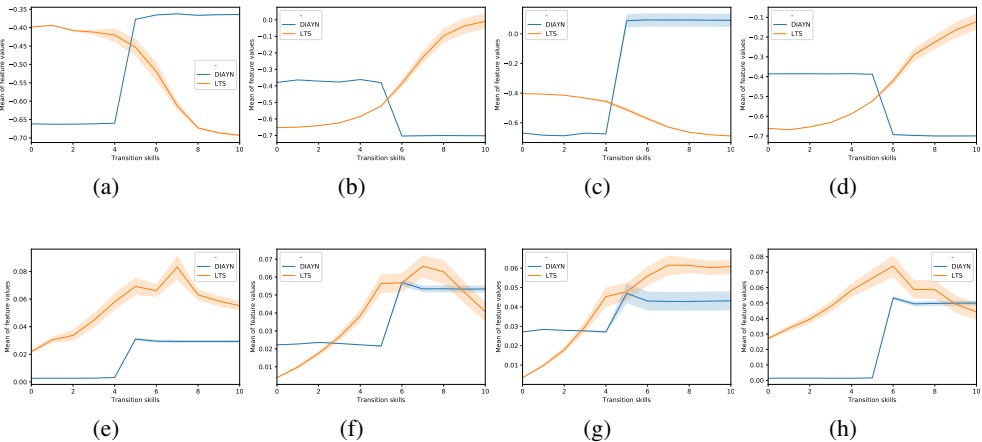

Figure 16: **Transitional skills comparison on MountainCar**. The subgraphs (a),(b),(c),(d) represent the mean of features in terms of the transitional skills, and subgraphs (e),(f),(g),(h) represent the variance of features.

