# OpenReview forum: "Learning transitional skills with intrinsic motivation"
_ICLR.cc/2020/Conference — Reject_

### Official Review · AnonReviewer3 · 2019-10-24
**Official Blind Review #3**

**Rating:** 3

**Review:**

General description
The paper tackles the problem of transition between different skills in hierarchical reinforcement learning.
In particular, they follow the work of VIC and DIAYN and define an information-theoretic objective function that maximizes the mutual information between the future states and options, given the initial state, while minimizing the distance between the options and the so-called transitional states. The algorithm, LTS, is compared to DIAYN on three environments, namely CartPole, MontainCar, and Pendulum.

General remarks,
The problem tackled and the proposed idea are interesting; however, I am not fully convinced by the derivation and the experiments of the paper. The writing is in general not particularly clear and the notations are hard to follow, the symbols are often bloated with superscripts that are not clearly defined, and mixing capital and small letters for random variables and their realization.

On the derivation, Eq 3. How is it possible to replace p(s^p | p^t) with p(omega|s_t)? I understand the connection between the two but what guarantees the mutual information is still maximized? (the whole derivation depends on that)

The experiments and the plots are interesting, showing a smoother transition between skills than DIAYN, however, it is still not clear how that can help solve the task at hand. Could the author elaborate on that please?

Some details:
- Page 2 \in should be \sim. In general, the notation does not clearly distinguish a random variable (in capital) from a realization (in small letters). For instance, page 3, big \Omega (the random variable I presume) is written with a subscript i to indicate the ith skill.
- Footnote 1 page 2, a log is missing in the MI definition.
- Page 3, what does t refer to in S^t_{i ,j, 1}?
- Page 3, muture -> mutual.
- the paper mentions the experiments are conducted on MuJoCo but the appendix mentions the classical OpenAI Gym experiments.

**Experience Assessment:**

I have published one or two papers in this area.

**Review Assessment: Checking Correctness Of Derivations And Theory:**

I assessed the sensibility of the derivations and theory.

**Review Assessment: Checking Correctness Of Experiments:**

I assessed the sensibility of the experiments.

**Review Assessment: Thoroughness In Paper Reading:**

I read the paper at least twice and used my best judgement in assessing the paper.

---

> ### Author Response · Authors · 2019-11-15
> **Response to Official Blind Review #3**
>
> We thank the reviewer for the thoughtful and constructive feedback. It has greatly helped us improve the paper, which involves the following details:
>
> 1. Q: The writing is in general not particularly clear and the notations are hard to follow, the symbols are often bloated with superscripts that are not clearly defined, and mixing capital and small letters for random variables and their realization.
>
> A: We have simplified the symbols and superscripts, and removed some unnecessary variables, such as $s^p_i$ and $S_{i,j,k}$. Please refer to the revised paper for more details.
>
> 2. Q: On the derivation, Eq 3. How is it possible to replace p(s^p | p^t) with p(omega|s_t)? I understand the connection between the two but what guarantees the mutual information is still maximized? (the whole derivation depends on that)
>
> A: We have revised $p(s^p | s_t)$ in Equations (3) and (4) into $p(\omega | s_t)$ according to this comment. In the revised paper, the process of learning skill $z_{i,j,k}$ directly depends on maximizing $I(S; \omega) + \alpha f^d$, where $f^d$ is a compensative term. Here, we give a intuitionistic interpretation: $E[log p(w_i|s_t) - \log  p(w_i)]$ increases and $f^d$ drops when $z_{i,j,k}$ is closed to $w_i$, otherwise the former one reduces and the latter one boosts. Hence, we maximize $E[log p(w_i|s_t) - \log  p(w_i)]$ to tradeoff information and distance in order to discover transitional skill $z_{i,j,k}$ between $w_i$ and $w_j$. We also revised the subsequent deduction. Please refer to the revised paper for details.
>
> 3. Q: The experiments and the plots are interesting, showing a smoother transition between skills than DIAYN, however, it is still not clear how that can help solve the task at hand. Could the author elaborate on that please?
>
> A: In our tasks (CartPole, MountainCar, PenDulum and HalfCheetah), the high-level meta control always choose primitive skills to accomplish complex tasks. In the baseline DIAYN, the primitive skills are trained to be distinguishable. When the agent reaches a state that is uncorrelated to the next skill, it may fail to execute the next skill to accomplish the task. The introduced ‘transitional skills’ enable the agent to transit from one to the next primitive skills, thus all the skills can be executed to accomplish complex tasks. Hence, our proposed algorithm performs better than DIAYN. Please refer to the following Table 1 for details.
>
> Table 1.Success rate(%) for practical tasks.
> ----------------------------------------------------------------------------
> 					DIAYN			LTS
> 		Policy    weighted random master 	weighted random master
> ----------------------------------------------------------------------------
> 		MountainCar	 0	 	0	 	20.8	 	0.2	 50.2	 100
> 		Pendulum	 0	 	14.2	 	100	 	1	 100	 100
> 		CartPole	 	1.6	 	10	 	36.8	 	0	 25.8	 48
> 		HalfCheetah	 0	 	29.2	 	44.7	 	0	 43.0	 78.6
> ----------------------------------------------------------------------------
>
> 4. Q: Some details need to be revised:
> - Page 2 \in should be \sim. In general, the notation does not clearly distinguish a random variable (in capital) from a realization (in small letters). For instance, page 3, big \Omega (the random variable I presume) is written with a subscript i to indicate the ith skill.
> - Footnote 1 page 2, a log is missing in the MI definition.
> - Page 3, what does t refer to in S^t_{i ,j, 1}?
> - Page 3, muture -> mutual.
> - the paper mentions the experiments are conducted on MuJoCo but the appendix mentions the classical OpenAI Gym experiments.
>
> A: We have revised the following details according to your comments:
>
> (1) Revising $\omega \in p(\omega)$ to $\omega \sim p(\omega)$.
> (2) Mending the expression of mutual information in footnote 1 on page 2.
> (3) Removing $S^t_{i,j,1}$ on page 3.
> (4) Mending 'muture' to 'mutual'.
> (5) We are sorry to admit that our previous experiments are irrelevent to MoJoCo. Hence, we have revised our mention into 'The tasks of CartPole, MountainCar, Pendulum and HalfCheetah-v3 are based on OpenAI gym \footnote{http://gym.openai.com/}'.
>
> Moreover, we have added supplementary experiments in the revised paper. For more details about our experiments, please visit our site for video demonstrations: https://sites.google.com/view/lts-skill

---

### Official Review · AnonReviewer2 · 2019-10-24
**Official Blind Review #2**

**Rating:** 6

**Review:**

The paper aims to learn transitional skills in addition to creating diverse primitive skills without a reward function by using an information theoretic objective, borrowing ideas from instrinsic motivation. The paper addresses the challenge of learning diverse primitives and also learning to transition between any pair of them, which is valuable for real-world complex problem solving with sparse rewards. The proposed LTS approach is compared againt a state-of-the-art DIAYN approach on several benchmark control tasks.

I have several comments/concerns regarding the experimental results.

1. How would the feature engineering used (Appendix E) affect the performance of tasks solved using the primitive skils learned using LTS and how `would policies learned with this framework perform compared to standard RL policies learned individually with external rewards? This is to establish whether the learned skills and transitions between them are useful for any down-stream tasks. In the DIAYN paper, experiments on down-stream tasks were provided.

2. What are these learned primitive tasks for each benchmark task? A visualization (or some interpretation of the meaning of the skills learned) would prove useful in understanding the primitive skills learned. Again, the DIAYN paper visualizes the primitive skills learned in their tasks.

3. I don't understand how the results in 6.3 demonstrate the generalization ability of the method. The authors clearly state that using 3 transtional skills fails when evaluated on 50 transitional skills. Can this be elaborated on?

4. Related to 1 above, I found it difficult to relate the features used for the tasks with the actual tasks themselves. More background information and motivation for the features is required (as opposed to saying it makes distinguishing between skills better -- what does better mean here?).

5. Lastly, the paper is full of silly typos, such as the incorrect usage of singular and plural nouns.

**Experience Assessment:**

I do not know much about this area.

**Review Assessment: Checking Correctness Of Derivations And Theory:**

I assessed the sensibility of the derivations and theory.

**Review Assessment: Checking Correctness Of Experiments:**

I assessed the sensibility of the experiments.

**Review Assessment: Thoroughness In Paper Reading:**

I read the paper at least twice and used my best judgement in assessing the paper.

---

> ### Author Response · Authors · 2019-11-15
> **Response to Official Blind Review #2**
>
> Thank you for your thoughtful review and questions, we very much appreciate the time you took to review our work. We reply to your points below.
>
> 1. Qs: (1)How would the feature engineering used in Appendix E affects the performance? (2)How would polices learned perform compared to standard RL? (3)Whether the learned skills are useful for down-stream tasks?
> As: (1)Here we note that the feature value and its statistical characteristics is just used to visualize the trajectories and effectiveness of skill transition (i. e. show the performance). The states used to discriminate the skills by our discriminator are the whole states of the agent instead the single feature value. Hence, the feature value never affects the performance of our proposed algorithm.
> (2)The standard RL algorithm is less effective to accomplish the tasks in this paper (such as the acrobatics of half-cheetah), where the comparison with standard RL can be achieved in our baseline DIAYN. In standard RL, the update of value function is based on a dedicated extrinsic reward , however, in some complex tasks, such as the acrobatics of half-cheetah, it is nearly impossible to design a perfect reward function. While in our setting, we need the agent to explore the environment learning primitive skills, then adopting a meta-policy to choose these low-level skills. In our work, we focus on the transition process of different primitive skills instead of collecting rewards from the environment.
> (3)The learned skills are useful for down-stream tasks. In the revised paper, we add supplementary experiments of down-stream tasks to show the effectiveness of skills. In the HalfCheetah task, the agent is able to execute several specifical movement thanks to learning different skills. For more details, please refer to Section 6.5 (Transition with Hierarchical Framework) in the revised paper.
>
> 2.Q: What are these learned primitive skills for each benchmark task?
> A: Since the skills of CartPole, MountainCar and Pendulum are hard to visualize and most of the skills are actually periodic movement (subject to the environment), we utilize the statistical characteristics of feature value to represent different skills, such as the angle and the position. In our revised paper, we have added supplementary experiments of the HalfCheetah task, and illustrated the primitive skills in Figure 2(c) and the transition trajectories in Figure 3(c). The agent in HalfCheetah task has high dimensional state and action space, the success skill transition based on our work indicates the superiority the proposed algorithm. For more details, please refer to the revised paper, or visit our site for video demonstrations: https://sites.google.com/view/lts-skill
>
> 3.Q: The generalization ability of the method.
> A: In the experiment, we set the number of transition skill K-1=3, and the set of the transitional skills index z is {0.25, 0.5, 0.75}. We use the three transitional skills to the policy \pi (a_t | s_t, z), and train the network. While we test 50 transitional skills based on the model trained before, and the feature value shows a smooth transition between different primitive skills. Then, we further reduce the step of transitional skills, the feature value still shows smooth transition, which indicates the strong generalization ablitiy of our proposed algorithm. Plus, the strong generalization ability of our proposed algorithm can be utilized to reduce the training complexity.
>
> 4. Q: Is using features to distinguish skills?
> A: We are sorry for not having provided enough reasons for introducing feature value in the previous paper. Plus, we think we may not understand the question clearly and we try to dispel your concern. Actually, the feature value we used in our paper is just to visualize the diverse skills. And we use the whole state information to discriminate the skills, which is done by the discriminator. The 'better' we noted in Appendix E is that we hope to better show the difference of the trajectories visited by different skills. An example is that the diverse skills may learn different position while the same velocity, e.g. skill \omega_1 goes right and skill \omega_2 goes left. We can think these are two different diverse skills. So we visualize the different skills with the position.
>
>
> 5.Q: There exists lots of typos.
> A: Thank you for your reviews. We are sorry to admit that there do exit lots of typos in the previous paper. Hence, we have revised and updated our paper, changing the typos, description and adding a new experiment for further analysis. Please let us know if you have any other questions or concerns.

---

### Official Review · AnonReviewer1 · 2019-10-24
**Official Blind Review #1**

**Rating:** 3

**Review:**

What is the specific question/problem tackled by the paper?
This paper addresses the problem of learning and using useful skills through unsupervised RL. While past work (VIC, DIAYN) found skills by maximizing the mutual information between skills and states, this paper (LTS) aims to find transitional skills to move between the primitive skills.

The proposed method is, if the primitive skills found by DIAYN are labeled w_i, to find skills z_[i,j,k] which corresponds to the kth state while transitioning from w_i to w_j. These transition skills are learned jointly with the primitive skills using the mutual information objective.

The results in the paper are on mountain car, cart pole, and pendulum. The plots show that distinct primitive skills can be learned in these environments, and that the transition skill transitions between them. The results do not show the actual performance of using these skills for any specific task, which was the motivation presented in the introduction. The

Is the approach well motivated?
The approach is not well motivated. The paper claims that transitional skills are necessary, but does not show any evidence of an agents behavior improving by using transitional skills.

I argue to reject the paper in its current state; Sections 3 and 4 need to be rewritten to be more easily understandable, and the need for transitional skills should be more empirically motivated.


Clarity Issues:
- The method is presented in a very confusing way. I am not confident that I understand how the transitional skills are defined.
- Figure 2: What is the “statistical characteristic” and “feature value”? The vague terms are not defined. Which value? What statistic?
- Figure 5: I see that LTS transitions more smoothly between skills, but why is DIAYN’s fast transition bad? Clearly it still succeeds at reaching the other state.

--------------------------------
Update
Due to the additional experiments that show the benefit of LTS, I have increased my score to weak reject.
Section 6.5: Thank you for adding this additional experiment. This section lacks details about how the experiment was run. How was the meta-policy trained? In “3) LTS-master: Transfer policy with optimal weights learned by meta-policy”, why is there transfer? Transfer from what to what?
“Moreover, we conduct hierarchical framework to weight (or choose) the action modeled by different primitive skills.” What does this sentence mean?

Reading over the paper again, the paper needs several editing passes to fix the grammar, as a number of sections (like above) are not understandable. While the paper has improved and is on a good track, I do not think it is yet ready for publication.

**Experience Assessment:**

I have published in this field for several years.

**Review Assessment: Checking Correctness Of Derivations And Theory:**

I assessed the sensibility of the derivations and theory.

**Review Assessment: Checking Correctness Of Experiments:**

I carefully checked the experiments.

**Review Assessment: Thoroughness In Paper Reading:**

I read the paper thoroughly.

---

> ### Author Response · Authors · 2019-11-15
> **Response to Official Blind Review #1**
>
> Thanks for your comments and questions. We very much appreciate the time you took to review our work. We reply to your points below.
>
> 1. Q: Lack of experiments for specific tasks.
>
> A: We are sorry to admit that we have failed to test our proposed algorithm based on complex tasks in our previous paper. However, we have added supplementary experiments in the revised paper, where the experiments involve complex tasks where the transition of skill in high dimensional state and action space is necessary.
>
> Specifically, in the supplementary experiments, we use the proposed algorithm to finish some complex tasks, such as making the ‘pendulum’ move right and left periodically or to enable the ‘half-cheetah’ to complete some acrobatics, where the states and actions are 17-dimensional and 6-dimensional, respectively. These complex tasks need the agent to transit among multiple skills. Our proposed algorithm to learn transition skills can always perform better than the baseline (DIAYN and Weighed DIAYN). For more details, please refer to Section 6.5 in the revised paper, or visit our site for video demonstrations: https://sites.google.com/view/lts-skill
>
> 2. Q: The motivation of the work.
> A: The proposed algorithm in our work can be applied into Hierarchical Reinforment learning (HRL).  In HRL, the high-level meta control always choose primitive skills  to accomplish complex tasks. In the baseline DIAYN, the primitive skills are trained to be distinguishable. When the agent reaches a state that is uncorrelated to the next skill, it may fail to execute the next skill to accomplish the task.  To solve this problem, we introduce ‘transitional skills’ to enable the agent to transit from one to the next primitive skills, thus all the skills can be executed to accomplish complex tasks.
>
> 3. Q: What is the definition of transitional skills?
> A:  According to DIAYN (Eysenbach et al. (2018)), primitive skills can be learned without external rewards. However, the primitive skills are independent with each other, and the agent can hardly transferred from one to other to accomplish complex tasks. Transitional skills denote the skills which can enable the agent to transfer from one primitive skill to another smoothly. For more details, please refer to Sections 3 and 6 in the revised paper.
>
>
> 4. Q: In Figure 2,what is the feature value and the statistical characteristics?
> A: In Figure 2(a), we adopt feature values to visualize the states of agent, which are ‘Pole Angle’, ‘Position’ and ‘Sin(Angle)’ for CartPole, MountainCar and PenDulum, respectively. The statistical characteristics in Figure 2(b) corresponds to the trajectories of feature values in Figure 2(a) (the ‘Pole Angle’, ‘Position’ and ‘Sin(Angle)’). For more details about feature values, please refer to Appendix E.
>
> 5. Q: In Figure 5, does DIAYN have faster transition?
> A: We are sorry to admit that there exits a misleading in Figure 5, and we have modified the interpretation of Figure 5 in the revised paper. In fact, the value of the y-axis denotes the mean of feature values (which is also the statistical characteristic),  whose sharp fluctuation illustrates  large probabilities of failure of skill transition. For DIAYN's primitive skills, there exists a sharp leap which notes harder transition between the two skills, compared with the smooth line of LTS's transition. Hence, the sharp leap of DIAYN does not mean it has fast skill transition.
>
> Moreover, we have rewrite and update our paper, adding more experiments for further analysis.

---

### Decision · Program_Chairs · 2019-12-19

**Decision:**

Reject

**Comment:**

The submission has two issues, identified by the reviewers; (1) the description of the proposed method was found to be confusing at times and could be improved, and (2) the proposed transitional skills were not well motivated/justified as a solution to the problem the authors propose to solve.